# Techno-Economic Analysis of the Multiple-Pass Ultrasonication with Mechanical Homogenization (MPUMH) Processing of Processed Carrot Discards to Puree

**DOI:** 10.3390/foods12010157

**Published:** 2022-12-28

**Authors:** Gagan Jyot Kaur, Deepak Kumar, Valerie Orsat, Ashutosh Singh

**Affiliations:** 1School of Engineering, University of Guelph, Guelph, ON N1G 2W1, Canada; 2Department of Chemical Engineering, SUNY College of Environmental Science and Forestry, Syracuse, NY 13210, USA; 3Department of Bioresource Engineering, McGill University, Sainte-Anne-De-Bellevue, QC H9X 3V9, Canada

**Keywords:** techno-economic analysis, carrot rejects and waste, processed carrot discards, carrot puree, ultrasonication

## Abstract

A sustainable valorization process for puree processing from processed carrot discards (PDCs) was proposed by using multiple-pass ultrasonication with mechanical homogenization (MPUMH), optimized at 9 min ultrasonication followed by mechanical homogenization for 1 min, subjected to three passes. Techno-economic analysis of the puree processing plant was studied for two process models using SuperPro Designer for a plant with a capacity of 17.4 MT/day, operational for 26 weeks, with a 20-year lifetime. The two scenarios were (i) base case (PDCs processed without peels and crowns) and (ii) case 2 (PDCs and carrots (50:50, *w/w*) processed with peels and crowns). Both scenarios were economically feasible with an internal rate of return (IRR) and return on investment (ROI) at 24.71% and 31.04% (base case) and 86.11% and 119.87% (case 2), respectively. Case 2 had a higher total capital investment (Can$13.7 million) but a lower annual operating cost (Can$8.9 million), resulting in greater revenue generation (Can$29.7 million), thus offering a higher ROI. Sensitivity analysis related to the number of passes on puree quality and price is suggested to lower the capital investment. For the base case, a lower ROI was due to the high labor cost incurred for manual peeling of PDCs, indicating the critical need for developing a commercial peeler equipped to cut labor costs and increase profitability. The study casts insights into the techno-economic performance of a sustainable process for the valorization of PDCs.

## 1. Introduction

The world population is estimated to increase from 6.3 billion to 9.1 billion by 2050. An estimated 70% increase in food production will be required to meet the food demand of nine billion people [1]. Climate change, increasing land use for non-food crop production and limited availability of water resources have compelled researchers to exploit other potential options to fulfill the predicted future food needs of the growing population. Cultivation of high-yielding crops and reduction in food losses are among the potential possible solutions. Every year, approximately 1.3 billion tons (one-third of the total production) of food is lost globally in post-harvest operations [2]. Among the agricultural produce (from both plant and animal origin), the highest percentage of losses (44%) are recorded in fruits and vegetables [3].

Carrots (Daucus *carota* L.) are among the top seven vegetables produced in the world, with a production of 40.95 million metric tons (MMT) in 2020 [4]. It is also among the top horticultural crops in Canada, contributing Can$129.882 million dollars to the country’s economy [5]. Carrots are consumed both as fresh vegetables in salads or as sides, and in processed food, in dressings, stews, soups and juices. It is estimated that 20–30% of carrots are separated as carrot rejects and waste (CRW) during the primary processing of the carrots comprising oversized, undersized, and processed carrot discards (PDCs). Among these PDCs are the major underutilized fraction of CRW [6], due to non-uniformity in size and shape, and the presence of undesirable bitter components in the peel and petioles attached to the crowns. Their utilization is limited to partial use in animal feed [7] and for biogas production using an anaerobic digestion process [8]. A significant fraction of PDCs ends up in landfills, causing environmental deterioration by generating greenhouse gases (GHG). In addition to these direct emissions, there are significant indirect emissions associated with food waste, if we consider the water and energy used to produce the lost food [2]. Our previous studies [9] demonstrated the conversion of PDCs to carrot puree using conventional thermal treatments. The study revealed that exposure to high temperatures during conventional thermal processing possibly caused isomerization of the β-carotene, thus resulting in its lower percentage in carrot puree.

In the latest study [10], we developed and optimized a novel method, i.e., multiple-pass ultrasonication with mechanical homogenization (MPUMH), for the processing of carrots to puree. This novel method replaced the conventional thermal treatment with ultrasonication followed by homogenization, which allowed 80% retention of β-carotene and fiber content, thus enhancing the nutritional value of the processed product. Similarly, encouraging results were reported on MPUMH processing of PDCs to carrot puree [11], capable of reducing the potential landfill load, thus reducing the GHG and contributing towards future food security. The process appears to be attractive on the laboratory scale; however, before making scale-up efforts and realizing the practical aspects of the technology, it is important to determine the commercial scale economic viability of the process. Techno-economic analysis using process simulation models is an effective tool to determine the commercial-scale economic matrices of new and early stage technologies [12,13,14]. In addition to determining system economics, the process models help in conducting comprehensive mass balance and identifying critical processes and parameters, and their effect on the overall profitability of a process. A significant number of recently published studies on the techno-economic analyses (TEA) of food waste valorization are focused on bioenergy/biorefinery processes, namely hydrothermal oxidation for the production of hydrochar [15], biodiesel production [16], power generation from biogas [17], ethanol production by enzymatic hydrolysis and fermentation [18], production of lactic acid, lactide and poly(lactic acid) [19], plasticizer, lactic acid and animal feed [20], polyhydroxyalkonates and biofuels [21], and co-pyrolysis [22]. The results of these published studies indicate economic gains, offering sustainable solutions to the existing food waste management and utilization challenges. As per the author’s knowledge, limited TEA studies are reported on the utilization of food rejects/overflow in food processing and food product development. The objective of this study was to develop a process model and perform a commercial-scale economic and technical feasibility analysis of the novel MPUMH carrot puree production process.

## 2. Materials and Methods

### 2.1. Process Model Development and Simulations

Comprehensive process models for the puree-making process were developed using SuperPro Designer V9.5 (Intelligen, Inc., Scotch Plains, NJ, USA), a platform commonly used to develop models of food and bioprocessing technologies. The models were developed for a capacity of 17.4 MT/day PCD processing, assuming 26 weeks (~6 months) of operation annually. The processing capacity was estimated based on the data from the large-scale carrot processing facility. Based on a discussion with them, it was assumed that 20% of the harvested carrots are rejected as PDCs. Out of those, about 5% are infected and were not accounted into the model. The processing was estimated based on the remaining 95% of PDCs (Table 1).

Process models were developed for two cases: (i) base case—carrot puree production from PDCs (after removing the peels and crowns), and (ii) case 2—carrot puree production from the mix (PDCs: graded carrots, 50:50), processed without removing peels and crowns. The processing capacity was kept at 17.4 kg/day, the same in both cases, which implies that in case 2, only half of the available PDCs were processed. The commercial processing plants vary in size and scope, so the process simulation models were developed as generic plants including all major equipment and unit operations necessary for the overall processing operation. The overall process consisted of 4 major processing sections: material handling and washing, peeling and chopping, puree making, and sterilization and packaging. Flow diagrams of process models (from SuperPro Designer V9.5) are shown in Appendix A and Figure 1. Brief details of each section are provided below.

The moisture content of PDCs was 86% (wb). These were conveyed to the washing unit for cleaning using water to remove the dirt and other foreign particles, which led to a 2% mass loss. The assumption of a 2% loss was based on the actual data obtained from the carrot processing facilities. There was no significant moisture change during this step. The next step was the peeling of PDCs (for the base case). After discussion with several food processing companies, it was found that there is no efficient peeler available for peeling PDCs due to their uniqueness/non-uniformity in shape and size. So, the peeling was performed manually. The time required for the manual peeling of PDCs was one labor hour for 3–4 kg of PDCs. It was similar to the data reported for kiwi peeling [23]. Based on the processing capacity of the puree-making plant and assuming 3.5 kg PDCs peeling/labor-h, about 202 labor was estimated for the current facility. Since it was only a manual peeling process, unskilled labor was considered at an hourly wage of Can$12.5/h. Based on the data collected from the preliminary runs in the lab, the model was developed assuming 15.6% material loss during the peeling operation which included the removal of crowns also. The peeled PDCs were chopped into small sizes using a commercial food chopper/shredder. The chopped PDCs were hydrated for 5 min with an equal amount of water (1:1, *w/w*) and pre-processed into a puree using a mechanical homogenizer. Due to the increase in the water, the solid content at the end of mechanical homogenization was about 7%. This pre-processed puree was subjected to sequential ultrasonication (9 min) followed by mechanical homogenization (1 min) for three passes [10]. Since MPUMH eliminated the conventional pre-heating process, the exposure of PDCs to multiple passes followed by homogenization enables the particle size reduction of the carrot puree comparable to the commercially available carrot puree. Case 2 was different from the base case as it involved the processing of PDCs and graded carrots (50:50, *w/w*) without removing the peels and crowns (peeling process was eliminated) (Figure 1). Carrots peels are reported for bitter components (polyacetylenes) and carrots are reported for β-carotene. Bohlmann, (1967) [24,25] correlated carrot consumption and low cancer risk due to the coexistence of bitter compounds and β-carotene in carrots. Since PDCs had lower β-carotene [9], these were mixed with carrots (50:50) [11]. The processing conditions and ratios were selected based on the desirable characteristics of the processed product, viz. particle size distribution, color, and carotenoid content. In the final section, the processed purees were sterilized and packed in 150 g containers.

### 2.2. Itemized Cost Estimation

To carry out itemized cost estimation, each economic parameter was separated according to capital and operating costs. In this study, the total capital cost consists of direct fixed capital cost (DFC) and working capital. DFC accounts for equipment purchase costs (belt conveyor, shredder, grinder, ultrasonicator, heat sterilizer, and shredder), direct costs (includes installation, insulation, piping, electrical facilities, buildings, and auxiliary facilities), and indirect costs (includes engineering, construction, and contingency). In capital cost, the equipment cost for carrot puree production capacity was calculated to scale up based on the relation between the equipment cost and equipment attribute, as shown in Equation (1):(1)$$C₂=C₁*\;{\left(\frac{A₂}{A₁}\right)}^{n}$$
where C_1_ is reference equipment cost (Can$); C_2_ is the adjusted equipment cost (Can$); A_2_ is the current size; and A_1_ is the base (reference equipment) size.

Depending on the purpose of the study, these direct and indirect costs are determined in detail and set individually as a certain percentage of equipment purchase cost or are calculated using a Lang factor, a common approach used in the process models developed for research [13,14]. A Lang factor of 3 was summed in the current study, which means that these direct and indirect costs were estimated as 200% of the equipment purchase cost. The working capital was assumed as 5% of the DFC.

The annual operating costs are estimated by accounting for raw materials, labor, utilities, laboratory/quality control/quality assurance (QC/QA), coproducts (if any), and facility-dependent costs. Although PDCs are regarded as waste, a small price (Can$0.018/kg) was used in the process simulations to account for the material handling cost. The cost of electricity, process steam, and cooling water were assumed to be Can$0.075/kWh, Can$12/MT, and Can$0.07/MT, respectively. Two skilled operators were assumed to work in the puree processing section of the plant. In addition to the unskilled labor required for the peeling operation, two other unskilled workers were assumed to work in the material handling, sterilization, and packaging section (1 for each).

Since there is no co-product, the unit production cost was directly calculated by dividing the annual operational cost by the number of units (cans in this case) produced, as shown in Equation (2):(2)Unit puree production cost (Can$MP entity)=Total annual costAnnual production(Can$/yrMP entity/yr)
where MP is the flow of discrete entity “Filled entity” in the stream (carrot puree cans).

The return on investment (ROI) was calculated based on the operational costs, revenues, DFC, and income taxes.

### 2.3. Revenues

Carrot puree was the source to generate revenue from the proposed MPUMH processing of PDCs. Its selling price was assumed Can$0.60/MP entity.

### 2.4. Profitability Analysis

Profitability analysis was performed to assess economic profitability for the MPUMH process for processing puree from PDCs. In the study, economic performance parameters such as net present value (NPV) and internal rate of return (IRR) were obtained from a puree production capacity of 17.5 MT/day with a discounted rate of 20%, where NPV is the cumulative discounted cash flow value at the end of the project. Total project life was assumed to be 22 years, including the first 2 years for construction and set up. Total DFC was distributed over the first two years (40% and 60%, respectively). Income taxes were calculated as 35% of the taxable income. A modified accelerated cost recovery system (MACRS), which uses a double declining balance depreciation method and then switches to a straight line, for a high tax deduction, was applied for 7 years for the reported study.

## 3. Results and Discussion

### Itemized Cost Estimation

Process simulations were performed for detailed material and energy balance using above mentioned assumptions, and data were analyzed to estimate process economics. The overall process economics (total investment, annual operating cost, and revenue) and puree production are presented in Table 2.

The total capital investment was estimated at Can$10 million for the base case, with an annual puree production of 5227 MT. The capital investment was about 37% higher (Can$13.7 million) for case 2, processing a 50:50 mix of PDCs and sound carrots (Table 3). Although the amount of raw material input was the same in both cases, the cost of the raw material was higher for case 2 as it included the additional cost of graded carrots (Table 4).

Moreover, in case 2, peels and crowns (accounting for about 16% of material in the base case) were a part of the raw material, thus resulting in a higher capacity of processing equipment. Since more raw material was processed, the puree production was 18.4% higher in case 2. The puree can production was estimated at 42 million and 50 million (approx.) for the base case and case 2, respectively (Appendix A). Figure 2 illustrates the equipment cost breakdown among the four sections of the processing plant, for both cases. In both cases, more than 70% of the cost is associated with the equipment cost for puree production, which is due to the high cost of homogenizers and ultrasonicators.

The annual operational costs for the base case were estimated Can$23.6 million. The operating costs for a processing plant consist of raw material, labor, facility dependent, and utility costs and the breakdown among these items is illustrated in Figure 3 (for details, refer to Appendix A). Usually, the raw material accounts for more than 50% of the operational costs in the food and bioprocessing operations. However, in the current case, raw materials accounted for Can$2.1 million, (9% of the annual operating cost) for the base case. This was due to the usage of PDCs (almost zero price). The raw material cost for case 2 was estimated at Can$3.67 million, which was 70% higher than that of the base case, mainly due to the cost of the graded carrots (Can$0.73/kg) used as input in case 2 (Table 4). The labor cost of Can$16.6 million was the highest contributor (70%) of the total operational costs in the base case, mainly due to the significantly high labor cost associated with the peeling operation (Table 5). Since case 2 does not consist of peeling operation costs, the total labor cost was estimated at only Can$1.84 million, accounting for 20% of the total operational costs (Appendix A). The utility cost in the base case and case 2 were estimated at Can$0.22 and Can$0.26 million, respectively, mainly associated with electricity usage (Appendix A). Facility-dependent costs that include expenses related to depreciation, maintenance, and insurance, were estimated at Can$2.16 and Can$2.94 million, respectively for the base case and case 2, contributing 9% and 33% to the annual operational costs (Appendix A). As the facility costs are directly proportional to the equipment cost, the cost was found higher in case 2 compared to the base case (Figure 3).

Considering the puree selling price of Can$0.6 per can (150 g), the revenue from base case and case 2 was estimated at Can$25.1 and Can$29.7 million, respectively. The revenue was higher for case 2 due to relatively more puree production. Considering this revenue, capital costs, operational costs, and economic assumptions mentioned in the method section, the ROI and IRR were 31.05% and 24.71% (base case) and 119.87% and 86.11% (case 2) (Appendix A). Even with the higher capital investment, the ROI, IRR, and NPV values were significantly higher for case 2 (using PDCs and graded carrots, 50:50) mainly due to higher revenue generation and lower operational costs.

## 4. Conclusions

The commercial-scale economic and technical feasibility analysis of the carrot puree production process reveals the scope of potential adoption of the MPUMH processing technology at the industrial scale. The preliminary TEA results of NPV > 0 and ROI > IRR with shorter payback periods are encouraging for both the base case and case 2. Further studies are suggested on the development of commercial peelers to cut the high labor cost reported in the base case. Similarly, for case 2, three passes are recommended for puree processing, which needs further investigation to correlate the effect of the number of passes on the puree quality and price that could potentially help in understanding the trade-off and achieving process optimization for lower capital investment.

## Figures and Tables

**Figure 1 foods-12-00157-f001:**
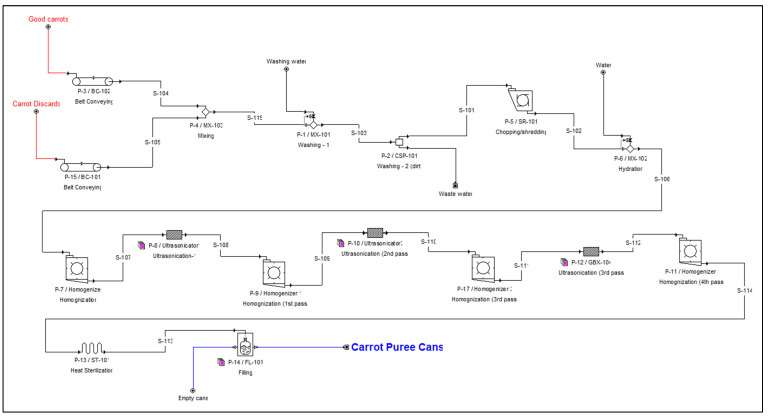
Flowsheet of a process model (case 2: mix of PDCs and graded carrots (50:50)) developed in SuperPro.

**Figure 2 foods-12-00157-f002:**
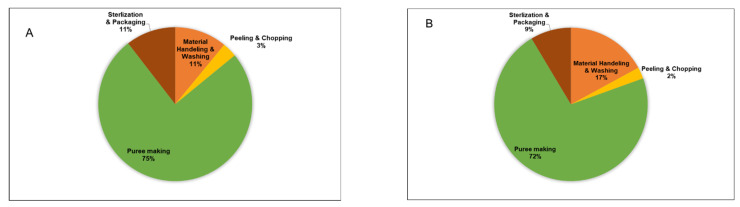
Equipment cost breakdown among various sections of the plant: (**A**) base case, (**B**) case 2.

**Figure 3 foods-12-00157-f003:**
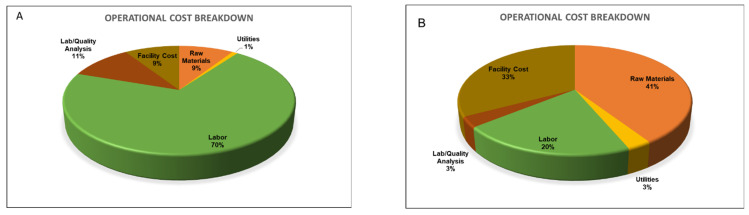
Operating cost breakdown for the carrot puree production plant: (**A**) base case and (**B**) case 2.

**Table 1 foods-12-00157-t001:** Estimation of the carrot rejects/waste generated in the Holland Marsh.

Details		Processing Capacity	Operational Weeks/Year	Capacity (kg)
Carrot processing units in the Holland Marsh	7			
Large scale units	3	900 pallets */week	26	16,732,170
Medium scale	3	600 pallets/week		-
Small scale	1	300 pallets/week		-
Loss variation	20–30%	20% (lower value)		3,346,434
Infected carrots	5%			167,457.9
Rejects safe for food processing				3,179,112.3
Capacity of plant/day				17,372.31

* Considering 1 pallet to be 715.05 kg.

**Table 2 foods-12-00157-t002:** Overall process economics of the puree production plant for two cases.

Cost	Base Case	Case 2
Total investment (000 Can$)	10,096	13,703
Annual operating cost (000 Can$/yr)	23,601	8978
Puree production (kg/yr)	5,227,746	6,193,850
Puree cans—150 g (000 cans/year)	41,822	49,550
Puree can unit production cost (Can$/can)	0.56	0.18
Revenue 000 Can$ (puree can @ Can$0.6/can)	25,093	29,730

**Table 3 foods-12-00157-t003:** Total capital investment in the (i) base case and (ii) case 2.

Description	Base Case (Can$)	Case 2 (Can$)	
Total equipment cost	3,205,000	4,350,000	
Installation cost	6,410,000	8,700,000	200% of total equipment cost
Total fixed capital cost	9,615,000	13,050,000	
Working capital	480,750	625,500	
Total capital investment	10,095,750	13,702,500	5% of total fixed capital cost

**Table 4 foods-12-00157-t004:** Details of the raw material cost for the (i) base case and (ii) case 2.

		Base Case		Case 2	
Description	Unit Cost (Can$)	Annual Rate	Annual Cost (Can$)	Annual Rate	Annual Cost (Can$)
Processed carrot discards (kg)	0.018	3,161,760	55,713.84	1,580,881	28,455.858
Empty cans	0.05	41,821,972	2,091,098.6	49,550,805	2,477,540.25
Water (MT)	1.25	4193	5242	4676	5845
Graded carrots (kg)	0.734			1,580,881	1,160,366.65
Raw material			2,152,054.44		3,672,207.76
			2.15 million		3.67 million

**Table 5 foods-12-00157-t005:** Annual labor demand and cost for the puree production plant (base case).

Labor Type	Annual Demand (Labor h/year)	Annual Amount (000 Can$)
Operator	17,472	1205.6
Unskilled labor (peeling)	1,179,360	14,742.0
Unskilled labor	11,648	174.7
Supervisor	4368	458.6

## Data Availability

Data is contained within the article or Appendix A.

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
