# Peer review of "Techno-Economic Analysis of the Multiple-Pass Ultrasonication with Mechanical Homogenization (MPUMH) Processing of Processed Carrot Discards to Puree"

_foods, 2022, doi:10.3390/foods12010157_

Round 1
Reviewer 1 Report
Review of the manuscript entitled ‘Techno-Economic Analysis of the Multiple-Pass Ultrasonication with Mechanical Homogenization (MPUMH) Processing of Processed Carrot Discards to Puree’
The present work aims to evaluate the feasibility of a sustainable valorisation process for puree processing from processed carrot discards (PCD’s) by exploring the use of multiple pass ultrasonication with mechanical homogenization (MPUMH). Moreover, two scenarios were conducted a base case (PCD’s processed without peels and crowns) and a case 2 (PCD’s and carrots (50:50, w/w) processed with peels and crowns). In my opinion, this study is interesting. Results are well presented, and the authors made the concept of the study clear enough. I recommend minor revision of this manuscript
- Materials and methods:
· Minor, line 112: Please clarify why there is a need of 3 passes. Also, is it possible to predict and comment the differences on the costs and feasibility of the process by maximizing or minimizing the passes?

Author Response
Response to reviewer’s comments
Comment: Minor, line 112: Please clarify why there is a need of 3 passes.
Response: Sincere thanks to the reviewers for their positive feedback. Regarding the three passes, it was observed from the experiments that the MPUMH eliminates the conventional pre-heating process, the exposure of PCD’s to multiple passes followed by homogenization enables the particle size reduction of the carrot puree comparable to the commercially available carrot puree.
Comment: Also, is it possible to predict and comment the differences on the costs and feasibility of the process by maximizing or minimizing the passes?
Response : Thank you for the suggestion. The feasibility and cost of the process in relation to the passes is beyond the scope of the present study but is suggested for the future research study. As mentioned in the above comment, the three-pass process produces carrot puree with quality comparable to the commercially available carrot puree. Although there would be some energy savings, however, the quality of puree will be affected negatively by reducing the passes. The detailed analysis would require a relationship between puree quality and its cost, which is not available at this point but would be considered in future studies.
Reviewer 2 Report
This manuscript reports on work that is highly relevant to the field of Foods. The Paper aimed to propose a valorization process for puree processing from processed carrot discards using modeling work. Also, TEa analysis was performed. Overall, the coverage and content of the manuscript are balanced, and both methodology and results are clearly described. The style is very readable. The introduction is informative and demonstrates the context of the proposed studies; it also explains why the determination of the commercial scale economic viability of the process is so important for the food processing industry. Materials and methods are clearly presented, and results are properly presented and discussed. Conclusions are based on the results obtained. My only remark is that units should be in SI system, not lbs (for example, Table 1). Now authors mixed. Also, costs are given in $, but it is unclear: USD or CAD. This should be clearly stated.
Author Response
Response to reviewer's comments
Comment: My only remark is that units should be in SI system, not lbs (for example, Table 1).
Response: The authors are thankful to the reviewers for their feedback. All the units are updated to SI units (Table1) and in the literature. All the changes are highlighted in blue/red font.
Comment: Now authors mixed. Also, costs are given in $, but it is unclear: USD or CAD. This should be clearly stated.
Response; All the costs are CAD$ and are updated in the manuscript to reflect the same. The changes are highlighted in blue/red font.
Reviewer 3 Report
Minor edit: replace "lab" with "laboratory" on line 60
Include version number of the SuperPro Designer software on line 72
Suggest adding the SuperPro design file as Supplementary Material. It may be useful to other SuperPro users.
Author Response
Response to reviewer’s comments
Comment: Minor edit: replace "lab" with "laboratory" on line 60
Response: The suggested change is made in the revised manuscript
Comment: Include version number of the SuperPro Designer software on line 72
Response: The suggested change is made in the revised manuscript
Comment: Suggest adding the SuperPro design file as Supplementary Material. It may be useful to other SuperPro users.
Response: As per the reviewer’s suggestions, we have provided the model file in the supplementary material